# Phase-Extraction-Based MFL Testing for Subsurface Defect in Ferromagnetic Steel Plate

**DOI:** 10.3390/s22093322

**Published:** 2022-04-26

**Authors:** Chengjian Ma, Yang Liu, Changyu Shen

**Affiliations:** China Jiliang University, Hangzhou 310018, China; 16605746666@163.com (C.M.); lymysoul@163.com (Y.L.)

**Keywords:** magnetic flux leakage, phase extraction, subsurface defect detection

## Abstract

Magnetic flux leakage (MFL) based on phase extraction for detecting the subsurface defects in ferromagnetic steel plate was investigated. The relationship between electromagnetic field phase and the subsurface defect was analyzed. Low-frequency alternating current (AC) excitation source and high-power magnetizer arrangement with Hall sensor were used to increase the skin depth of the MFL. Experiments results showed that 12 mm deep subsurface defect can be detected by using the phase extraction means, which is about two times higher than that by using the amplitude method.

## 1. Introduction

Magnetic flux leakage (MFL) testing technology [1,2,3,4,5] is widely used for defect detection in ferromagnetic material special equipment, i.e., the high-temperature boiler and high-pressure pipeline. The usual defect detecting method is focused on the signal’s amplitude of spatial magnetic field intensity [6,7]. Compared to no defect, the existence of the defect will influence the distribution of spatial magnetic field distribution, which means the magnetic field intensity above the defect will enhance and MFL will emerge [8,9,10]. Many theories were proposed to describe the mechanism of the MFL but most of them are so qualitative that they simply regarded this intensity enhancement as the leakage of the magnetic flux density from the inside of the ferromagnetic equipment [11,12,13,14]. Many studies used magnetic resistance or magnetic path model to describe the spatial magnetic field distribution while they subsequently used finite element method (FEM) to calculate specific magnetic field intensity instead [15,16,17], but most of them neglected the rigorous theoretical analyses. To further ascertain the principle of the magnetic flux leakage, Sun carried out a series of research and proposed a magnetic compression effect (MCE) theory [18,19], which indicates that the non-zero background magnetic field exerts a compression effect on the magnetic diffusion from ferromagnetic material to the air. Hence, MFL can be deemed as the joint effect of magnetic diffusion, magnetic refraction, and magnetic compression.

However, MFL testing technology based on signal amplitude performs poorly in detecting the subsurface defect in ferromagnetic material [20,21]. It is well known that electromagnetic wave attenuates gradually in the conductive material. Since the ferromagnetic material is usually conductive, deep subsurface defect in the ferromagnetic material will cause weak MFL response. Hence, the defect’s magnetic signal is almost the same with the situation of no defect.

In this paper, the MFL testing technology based on signal phase was proposed for subsurface defect detection and a comprehensive theoretical analysis was presented. An MFL testing configuration was designed on the bias of the theoretical analysis. A comparative test was conducted to demonstrate the superior performance of the phase difference method.

## 2. Theoretical Analysis

The low-frequency MFL testing model proposed in this letter is illustrated in Figure 1. Alternating currents (ACs) are applied in the excitation coils which are encircled onto the magnetic yoke. A Hall sensor is used to collect the spatial magnetic field signal. Since the electromagnetic wave energy attenuates gradually inside the lossy medium [22,23], it is not easy to detect deep subsurface defects inside the sample steel. Generally, the electromagnetic wave energy mainly distributes at the “skin” of the medium. Hence, finding some ways to increase the skin depth *δ* is undoubtedly beneficial for deep subsurface defect detection.

The skin depth *δ* of the magnetic field in the sample steel has the following expression [20,22],
(1)$$\delta =\sqrt{\frac{2}{\mu \sigma \omega }}\;$$
where μ is the permeability, σ is the conductivity, and ω is the angular frequency.

From Equation (1), we can know that *δ* will become large with the decrease of ω and μ. Hence, we use a low-frequency AC excitation source to induce a spatial electromagnetic wave and use a high-power amplifier to generate a strong magnetic field intensity H to make the steel plate (20# steel as an example) in a magnetic saturation state, as illustrated in Figure 1. With the increase of the magnetic field intensity H, the magnetic flux density B increases slowly and tends to be a steady value while relative permeability $\mu_{r}$ decreases gradually, as shown in Figure 2 [24]. When H reaches 15,000 A/m, $\mu_{r}\approx 100$.

Electromagnetic field distribution is illustrated in Figure 3. Space can be divided into three mediums, air–steel–air. The specific expressions of electromagnetic field in each medium are as shown in Table 1.

$E_{i}$ and $H_{i}$ represent incident electromagnetic wave, while $E_{r}$ and $H_{r}$ represent reflective electromagnetic wave. $\beta_{1}$, $\gamma_{2}$, and $\beta_{3}$ are propagation constants in three mediums, respectively. *d* is the distance between the defect and the upper surface.

Since the tangential components of electric field E and magnetic field H are continuous at the interface of two mediums, E and H are satisfied for the following expressions:(2)$$\{\begin{array}{c} (E_{2i}+E_{2r}){|}_{z=d}=E_{3i}{|}_{z=d}\\ \left(H_{2i}+H_{2r}\right){|}_{z=d}=H_{3i}{|}_{z=d} \end{array}\Rightarrow R_{2}=\frac{\eta_{3}-\eta_{2}}{\eta_{3}+\eta_{2}}$$
(3)$$\{\begin{array}{c} (E_{1i}+E_{1r}){|}_{z=0}=(E_{2i}+E_{2r}){|}_{z=0}\\ \left(H_{1i}+H_{1r}\right){|}_{z=0}=(H_{2i}+H_{2r}){|}_{z=0} \end{array}\Rightarrow R_{1}=\frac{\eta_{eff}-\eta_{1}}{\eta_{eff}+\eta_{1}}\;$$
where $R_{i}$ is reflection coefficient of electric field and $\eta_{i}$ is wave impedance in medium *I* (*i* = 1, 2, 3). Since medium 1 and medium 3 are air, one can obtain $\eta_{1}=\eta_{3}=120\;\pi$. The effective wave impedances $\eta_{eff}$ of medium 2 and medium 3 are as follows:(4)$$\eta_{eff}=\eta_{2}\frac{\eta_{3}+\eta_{2}\tanh\left(\gamma_{2}d\right)}{\eta_{2}+\eta_{3}\tanh\left(\gamma_{2}d\right)}$$
and the complex propagation constant $\gamma_{2}$ can be expressed as
(5)$$\{\begin{array}{c} \gamma_{2}=j\omega \sqrt{\mu_{2}\varepsilon_{2}}\\ \varepsilon_{2}=\varepsilon -j\frac{\sigma_{2}}{\omega } \end{array}\Rightarrow \gamma_{2}=j\omega \sqrt{\mu_{2}\varepsilon \left(1-j\frac{\sigma_{2}}{\omega \varepsilon }\right)}$$

Because $\sigma_{2}$ is on the order of $10^{6}\;S/m$ and ε is on the order of $10^{-12}\;F/m$, one can obtain
(6)$$\frac{\sigma_{2}}{\omega \varepsilon }\gg 1\Rightarrow \{\begin{array}{c} \gamma_{2}\approx \sqrt{j\omega \mu_{2}\sigma_{2}}\\ \eta_{2}=\sqrt{\frac{\mu_{2}}{\varepsilon_{2}}}\approx \sqrt{\frac{j\omega \mu_{2}}{\sigma_{2}}} \end{array}$$

The final electromagnetic field can be deemed as the joint effect of incident wave and reflective wave,
(7)$$E_{1}=E_{1i}+E_{1r}=e_{x}E_{1im}\left[\left(1+R_{1}\right)e^{-j\beta_{1}z}+j2R_{1}sin\left(\beta_{1}z\right)\right]$$

Equation (7) indicates that the final electromagnetic field consists of a traveling wave field and a standing wave field. When the low-frequency excitation is used, then $\beta_{1}z\ll 1$. Hence, the electric and magnetic field expressions in medium 1 can be approximately written as
(8)$$\{\begin{array}{c} E_{1}\approx e_{x}E_{1im}\left(1+R_{1}\right)e^{-j\beta_{1}z}\\ H_{1}\approx e_{y}\frac{E_{1im}}{\eta_{1}}\left(1+R_{1}\right)e^{-j\beta_{1}z} \end{array}$$

Finally, the spatial magnetic field signal is obtained as follows:(9)$$H_{1}\left(z,t\right)\approx e_{y}\frac{E_{1im}}{\eta_{1}}\left(1+R_{1}\right)e^{-j\beta_{1}z}e^{j\omega t}=e_{y}H_{1m}e^{j\left(\omega t-\beta_{1}z+\varphi \right)}\;$$

The parameter φ can be expressed as
(10)$$\tan\varphi =\frac{im\left(1+R_{1}\right)}{\mathrm{re}\left(1+R_{1}\right)}$$

When steel plate has no defect, one can obtain
(11)$$\tan\varphi_{0}=\frac{im\left(1+R_{1}^{0}\right)}{\mathrm{re}\left(1+R_{1}^{0}\right)}=\frac{1}{1+\sqrt{\frac{2\omega \mu_{r}\varepsilon_{0}}{\sigma_{2}}}}\approx 1\Rightarrow \varphi_{0}\approx 45{}^{\circ}$$
where $\mu_{r}\;\mathrm{is}\;\mathrm{the}\;\mathrm{relative}\;\mathrm{permeability}\;\mathrm{in}\;\mathrm{steel}\;\mathrm{plate}\;\mathrm{and}\;R_{1}^{0}=\frac{\eta_{2}-\eta_{1}}{\eta_{2}+\eta_{1}}$. 

Then, the phase difference can be defined as follows:(12)$$\Delta \varphi =\left|\varphi -\varphi_{0}\right|\;$$

Although the expressions of electromagnetic field in Table 1 are approximate expressions, because the phases of electric field and magnetic field are not same and approximately have 90° delay when $\beta_{1}z\ll 1$ (low-frequency excitation), the above-mentioned deduction is still tenable since we use phase difference Δφ rather than phase φ as the detecting parameter.

Figure 4a shows the relationships between the phase of the MFL and the excited frequency and relative permeability $\mu_{r}$, respectively. The orange curve shows the change of phase φ and relative permeability *μ_r_* when the distance between the defect and the upper surface *d* is 10 mm and the excitation frequency *f* is 20 Hz; the mauve curve shows the change of phase φ and excitation frequency *f* when the distance between the defect and the upper surface *d* is 10 mm and the relative permeability *μ_r_* is 100. It can be seen that the optimization frequency for the MFL testing is lower than 60 Hz, and the optimization relative permeability $\mu_{r}$ is about 100. Furthermore, these optimization parameters also meet Equation (1) for increasing skin depth *δ*. Usually, the detection depth of the amplitude measurement method is less than 10 mm.

However, as shown in Figure 4b, here, by using the MFL testing technology based on the phase extraction method, the detection depth can be increased to 22.0 mm theoretically. For the purple region, the phase difference of the detection depth from 16.0 to 22.0 mm is smaller than that in the orange area, so it belongs to the uncertain region. Therefore, as the orange area shows in Figure 4b, the detectable depth is 16.0 mm.

## 3. Experimental Results and Discussions

Figure 5 shows the experimental system of the subsurface defects detecting based on the MFL testing. The system mainly comprises a signal generator, power amplifier, MFL sensor, data acquisition card, PC, and the specimen. The signal generator was used to generate the excitation signal (sinusoidal signal) which had a magnitude of 5 V and frequency of 20 Hz. The signal was amplified by a power amplifier (LM3886) and then fed to the excitation coil (with 300 turns) wrapped around both the right and left pole of the yoke. A Hall sensor, SS94A1, with the sensitivity of 25 mV/Gs was installed in the bottom of the MFL sensor to pick up the leakage flux signal. The Hall element was positioned equidistant between the poles of the yoke and configured to measure the tangential component of the MFL field with a lift-off distance of 1 mm. The detected signals were transmitted to the data acquisition card. The signals were finally transmitted to the PC through the data acquisition card.

Figure 6 shows the side view of the specimen. The specimen is made of ferromagnetic material (20# steel) with some artificial rectangular defect. The thickness of the specimen was 16 mm. The widths of those artificial rectangular defects were 3 mm. Artificial rectangular defects were uniformly distributed, and the distance between adjacent defects was 20 mm. We define the distance between the defect and the upper surface as the buried depth *d*, and their buried depths ranging from 3 mm to 15 mm in step of 1.5 mm (corresponding to the defects labels of No. 1 to No. 9, respectively).

In experiment, both of the amplitude and phase detection means were used to detect those defects, and the detection results were analyzed. Figure 7 shows the real-time signal of the leakage magnetic field at different buried depths by using the amplitude detection. It can be seen that when the buried depth *d* is lower than 6 mm, the amplitude of the signal is split into two peaks, and the corresponding defects can be monitored. The signal split means that the magnetic flux densities at the two edges of the defect were more intensive than other positions [24]. However, when the buried depth exceeded 6 mm, the amplitude of the signal almost remained unchanged (the voltage peak values showed the same value of 3.25 V), which means that one cannot detect those defects (the buried depths of 9 and 10.5 mm, for instance).

On the other hand, by using the phase difference extraction-based MFL testing, as shown in Figure 8, when the buried depth exceeded 6 mm, the phase difference peaks were different for variable buried depth, which indicated that the deeper subsurface defect could be monitored. Seven distinct peaks occurred and the phase difference value decreased with the increase of the subsurface defect’s depth *d*, as shown in Figure 8. Figure 9 shows the theoretical and experimental results for phase-difference-based MFL testing. During the experiment, due to the uneven movement speed of the sensor and the digitization of the sampling of the data acquisition card, the collected phase data will be disturbed and errors will be generated. The red points were the average value of all phases collected in multiple experiments for the defect at the same depth. The red ranges were the confidence interval with 95% confidence of phase value. The red line was the result of linear fitting of the red points. Through fitting calculation, the linear equation can be obtained as follows,
(13)y=−3.57x+51.46

Nevertheless, the phase differences corresponding to the defect’s depths of 13.5 mm and 15 mm are about 1.05° and 0.25°, respectively, which might be almost equivalent to the experimental error (caused by the measurement errors of the Hall sensor and the instability of the magnetic field). Therefore, here, the maximum depth of detection is 12 mm, which is about 2 times than the amplitude method. Moreover, the depth sensitivity of the phase difference testing method is about 3.6°/mm, which has excellent performance in term of depth sensitivity. The experimental results agree well with the theoretical analysis, further manifesting that phase difference is more sensitive than the signal amplitude in subsurface defect detection.

## 4. Conclusions

A method employing magnetic flux leakage (MFL) based on phase extraction was used to test the subsurface defects in a ferromagnetic steel plate. The relationship between electromagnetic field phase and the existence of the subsurface defect was analyzed. It was found that the phase difference was relevant to the buried depth. The advantages of the phase difference detection method are discussed through mathematical calculation. Theoretically, defects with a buried depth of 16 mm can be stably detected, and defects with a buried depth of 16 mm to 22 mm can be roughly detected. Experimental results agree well with the results of the theoretical analysis, and the results indicated that the superior performance of the MFL sensor based on phase difference was validated. The maximum depth of the phase difference testing method is 12 mm, which is about 2 times that of the amplitude method. The maximum depth sensitivity of 3.6°/mm was obtained, which showed excellent performance in term of depth sensitivity.

## Figures and Tables

**Figure 1 sensors-22-03322-f001:**
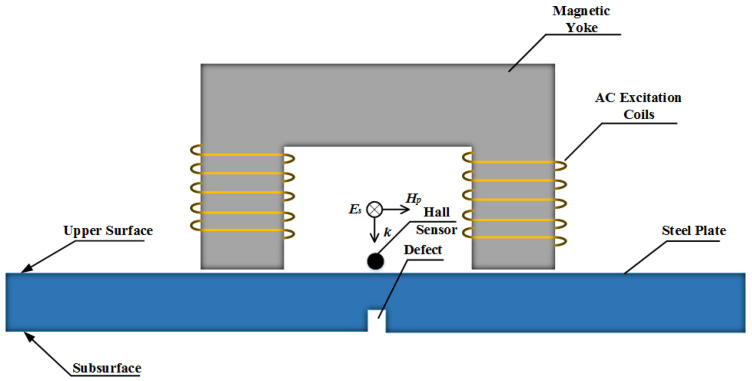
The proposed schematic of MFL measurement system.

**Figure 2 sensors-22-03322-f002:**
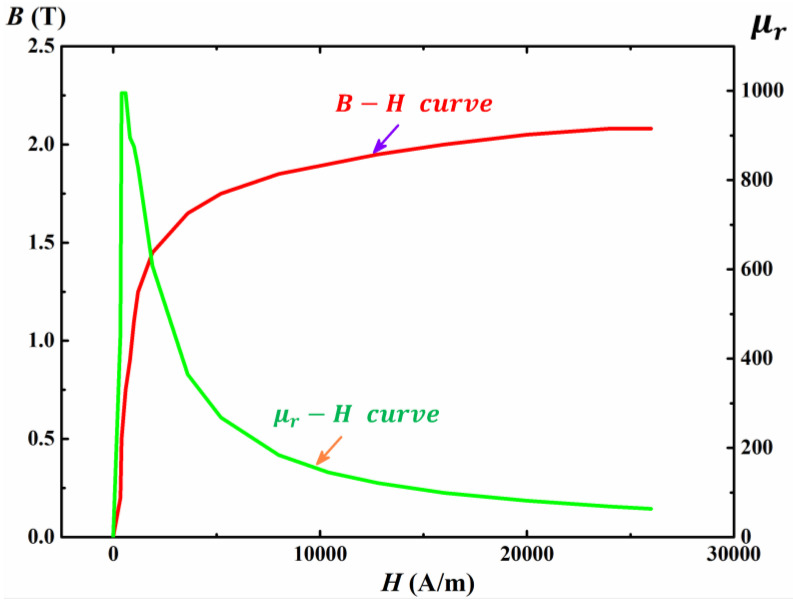
Magnetic characteristic of 20# steel.

**Figure 3 sensors-22-03322-f003:**
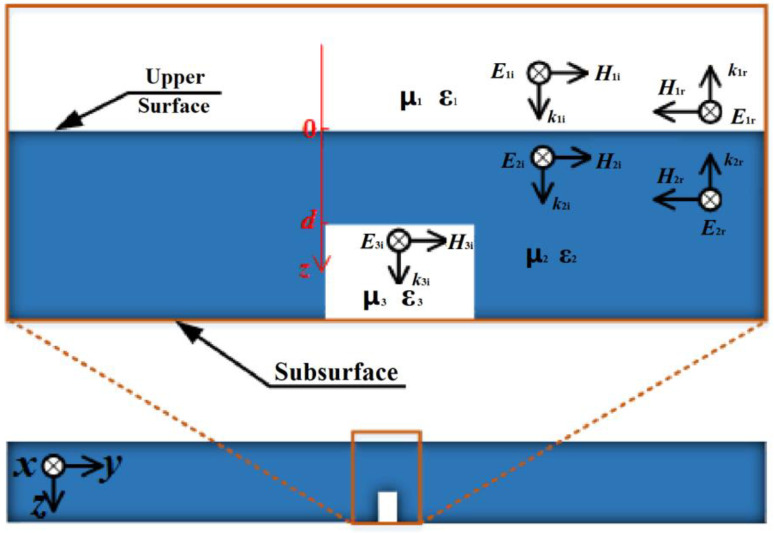
Electromagnetic field distribution.

**Figure 4 sensors-22-03322-f004:**
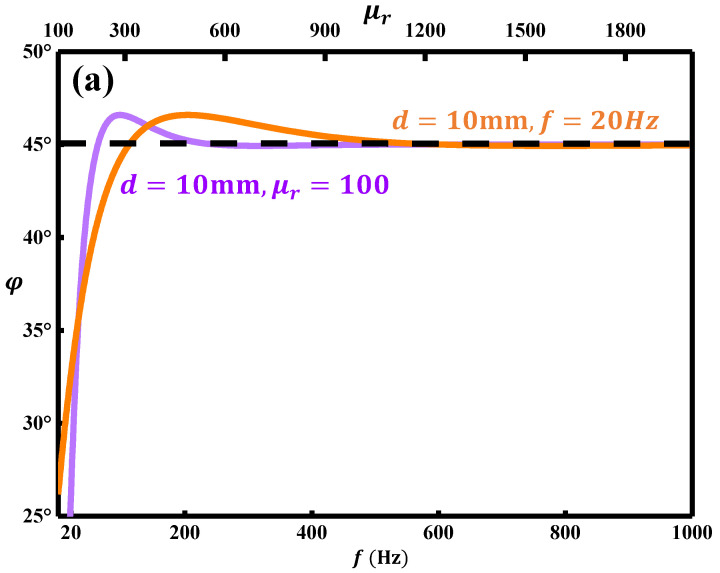
The phase φ with the change of (**a**) excitation frequency f and relative permeability $\mu_{r}$ of 20# steel; (**b**) subsurface defect’s depth d.

**Figure 5 sensors-22-03322-f005:**
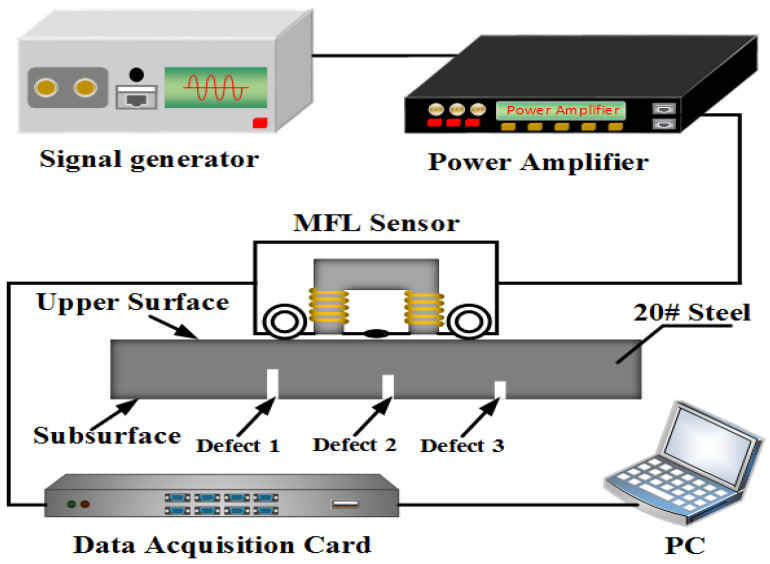
Experimental system of the subsurface defects detection based on the MFL testing.

**Figure 6 sensors-22-03322-f006:**
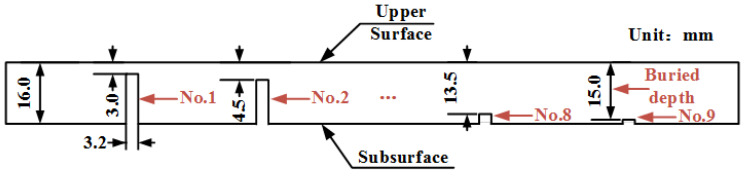
Geometrical sizes of defects in the specimen.

**Figure 7 sensors-22-03322-f007:**
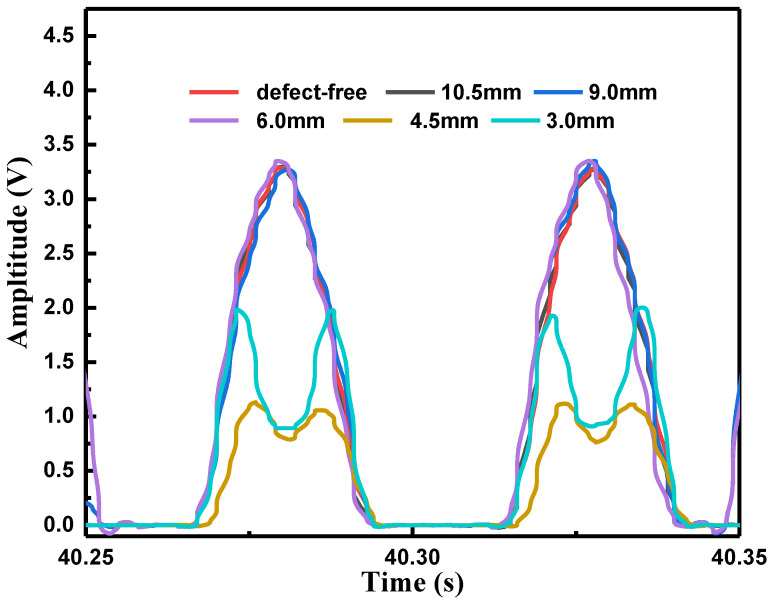
Real-time signals of the leakage magnetic field against different buried depths for amplitude detection.

**Figure 8 sensors-22-03322-f008:**
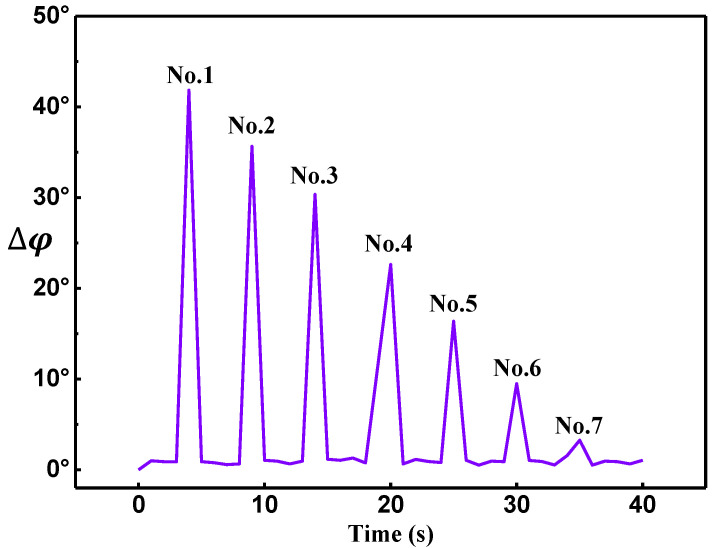
Phase differences for different buried depths.

**Figure 9 sensors-22-03322-f009:**
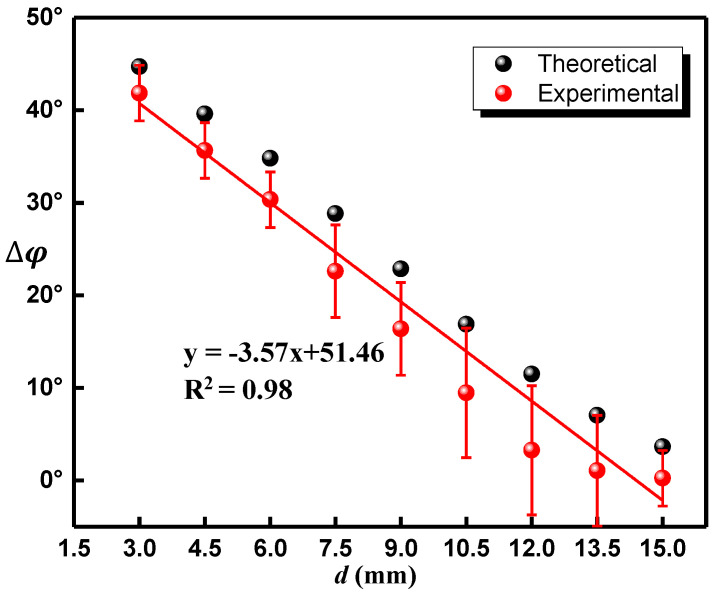
The relationship between the theoretical and experimental phase difference Δφ and d.

**Table 1 sensors-22-03322-t001:** Expressions of electromagnetic field.

$E_{1i}$	$H_{1i}$	$E_{2i}$	$H_{2i}$	$E_{3i}$	$H_{3i}$
$e_{x}E_{1im}e^{-j\beta_{1}z}$	$e_{y}H_{1im}e^{-j\beta_{1}z}$	$e_{x}E_{2im}e^{-\gamma_{2}\left(z-d\right)}$	$e_{y}H_{2im}e^{-\gamma_{2}\left(z-d\right)}$	$e_{x}E_{3im}e^{-j\beta_{3}z}$	$e_{y}H_{3im}e^{-j\beta_{3}z}$
$E_{1r}$	$H_{1r}$	$E_{2r}$	$H_{2r}$		
$e_{x}E_{1rm}e^{j\beta_{1}z}$	$-e_{y}H_{1rm}e^{j\beta_{1}z}$	$e_{x}E_{2rm}e^{\gamma_{2}\left(z-d\right)}$	$-e_{y}H_{2rm}e^{\gamma_{2}\left(z-d\right)}$		

## Data Availability

Not applicable.

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
