# Peer review of "Phase-Extraction-Based MFL Testing for Subsurface Defect in Ferromagnetic Steel Plate"

_sensors, 2022, doi:10.3390/s22093322_

Round 1

Reviewer 1 Report

Dear Authors, 

thank you for the presentation of the phase extraction based MFL testing for subsurface defect in ferromagnetic steel plate. 

I have three remarks:

1.State of the Art should be improved. In your References presented scientific materials are from years: 1999-2018. There is a need to add and review articles, scientific papers, patents last 5 years. Extend your paper including articles from years: 2018-2022.

2.Could you explain a quality of your research ? Please write more about errors in your experiments.

3. In Fig.9  you present the relationship between the theoretical and experimental results. In the experimental part: the red line, the red points and red ranges is a question about calculations of red ranges. How the red ranges were estimated?

Reviewer 2 Report

This paper proposed a method to evaluate the defect depth based on the phase extraction based on the MFL method, which is an interesting point for the readers. Besides, this paper is well organized. And the results show that this method can significantly improve defect recognition in a deeper subsurface.  So the reviewer recommends publishing this paper.

Author Response

Reviewer 2

This paper proposed a method to evaluate the defect depth based on the phase extraction based on the MFL method, which is an interesting point for the readers. Besides, this paper is well organized. And the results show that this method can significantly improve defect recognition in a deeper subsurface.  So the reviewer recommends publishing this paper.

Answer: Thank you very much for your positive comments.

Reviewer 3 Report

The Introduction is well-organized even it can be extended. Also, use the proper indication for the references with [1] and not as index. The quality of some images (for example 1 and 2) is poor. Please improved this aspect. Please indicate if the curves from figure 2 are original (measured by the authors, indicate the method, equipment) or taken from references (indicate a reference). The wave impedance is usually noted with Z not η. Equation (4) is original or must be cited? β1 parameter that appear from equation (7) is not define. Figure 4a contains two curves by some parameters are missing (f for the mauve curve and magnetic permeability for the orange curve). Also, please indicate the relevance of the colors in figure 4b. The experimental set-up presented in Chapter 3 do not indicate the placement of the defect? From all the following pictures, the place of the defect is not indicated. Please answer to this question. Also, the Conclusion chapter must be enlarged in order to emphasize the authors contributions. The references might be extended with more recent articles.

Round 2

Reviewer 3 Report

Thank you for considering all my remarks and answer all my questions. I have no other comments.